# Mixing Ratio and Cooling Rate Dependence of Molecular Compound Formation in OPO/POP Binary Mixture

**DOI:** 10.3390/molecules25225253

**Published:** 2020-11-11

**Authors:** Kengoh Nakanishi, Satoru Ueno

**Affiliations:** 1Miyoshi Oil & Fat Co. Ltd., Tokyo 124-8510, Japan; nakanishik@so.miyoshi-yushi.co.jp; 2Graduate School of Integrated Sciences for Life, Hiroshima University, Higashi-Hiroshima 739-8528, Japan

**Keywords:** binary mixtures of fats, molecular compound, POP, OPO, synchrotron radiation X-ray diffraction methods, palm oil

## Abstract

Owing to the increasing reports of the harmful effects of trans and saturated fatty acids, the demand for trans- and saturated-fatty-acid-free oil and fat products is increasing among consumers. However, it is difficult to maintain the product stability and shape retention of such oil and fat products. As a result, there is a high demand in the processed oil and fat industry to develop solutions to such problems. Herein, we used molecular compound (MC) crystals in an attempt to find alternatives to trans and saturated fatty acids. The MCs used were 1,3-dioleoyl-2-palmitoyl-*sn*-glycerol (OPO) and 1,3-dipalmitoyl-2-oleoyl-*sn*-glycerol (POP)—the main components of lard and palm oil, respectively. We believe that OPO and POP can be used to obtain no-trans, low-saturation, and high-oleic-acid oil and fat products. Optimal conditions for efficient MC crystallization were examined by changing the oil and fat composition under rapid cooling conditions assuming industrial cooling process by using differential scanning calorimetry and synchrotron radiation time-resolved X-ray diffraction methods. It was concluded that the increase in OPO concentration destabilized MC formation, while the increase in POP concentration stabilized it under rapid cooling conditions. As a result, it was shown that MC crystals can be efficiently obtained by reducing the degree of POP supercooling.

## 1. Introduction

Edible oil and fat products such as margarines and shortenings are manufactured by blending various types of oils and fats derived from vegetables and animals [1]. Triacylglycerol (TAG), in which three molecules of fatty acids are bound to glycerol, is their main component, and hence, their physical properties are greatly affected by the type of fatty acids in TAG [2]. The hardness of the edible oil and fat products is an important factor to consider during their manufacture. Edible oil and fat products containing a blend of oils and fats of various hardness values exhibit plasticity and consistency [3]. High-melting-point oils and fats are employed for producing crystal nuclei, medium-melting-point oils and fats for shape retention, and the low-melting ones for spreadability.

Further, to crystallize these oils and fats together, rapid cooling is usually performed using a scraped-surface heat exchanger during the manufacturing process [3]. The cooling rate is estimated to be ≥100 °C min^−1^ [4], and the products solidify when cooled. Although various physical changes such as shearing and pressure are observed when using a scraped-surface heat exchanger, we treated rapid cooling as the first step to clarify the crystallization behavior of edible oil and fat products.

The physical properties of these product change depending on the cooling rate and storage temperature, because oils and fats form crystal polymorphs [5,6]. Oils and fats typically form the following three types of polymorphs: α form for hexagonal crystals, β’ form for orthorhombic crystals, and β form for triclinic crystals. Furthermore, the γ form, which was observed in saturated-unsaturated-saturated mixed-acid TAG, and the sub-α form, which is more unstable and has a longer chain-length structure than the α form, have also been reported [2]. The lattice structure of the crystal polymorph was clarified by X-ray diffraction measurement. Table 1 summarizes the values of the long and short spacings in the typical polymorphs of oils and fats [4,7,8,9,10].

When the cooling rate of oil and fat products is increased, the degree of supercooling also increases, because the temperature decreases before the crystals could form. This leads to the nucleation of unstable polymorphs rather than the stable ones [11]. The more stable the polymorph, the higher the density, and the higher the melting point. Although the degree of supercooling is large, the energy barrier for polymorph formation is also large. Because of the competition between the two phenomena, the unstable polymorph crystallizes more quickly. Therefore, after melting the multicomponent edible oils and fats, it was observed that their crystallization behavior is significantly different when cooled under slow cooling conditions at room temperature from that observed under rapid industrial cooling conditions. Thus, the crystal polymorphs of each TAG constituting the oil and fat raw materials and the mixed composition of these polymorphs are greatly affected by cooling conditions [10].

A binary system of TAGs typically forms eutectic, monotectic, solid solution, and molecular compound (MC) phases [12,13]. The MC is formed by binding different molecules at a certain ratio. Its properties are different from those of their component molecules. For example, MC crystals in a binary system of 1,3-dioleoyl-2-palmitoyl-*sn*-glycerol (OPO) and 1,3-dipalmitoyl-2-oleoyl-*sn*-glycerol (POP), the main components of pork lard and palm oil, respectively, (1) can form a stable β polymorph without requiring any particular temperature control or long-term storage, (2) are solid at room temperature despite containing half of OPO in liquid form at room temperature, (3) contain a large amount of oleic acid moiety, which is good for health, and (4) have high storage stability [7,14,15]. In recent years, trans and saturated fatty acids have attracted interest because of their many health risks [16,17,18]. Therefore, oil and fat products should be crystallized using oils and fats rich in unsaturated fatty acids such as oleic acid.

If the oil and fat products contain a large amount of unsaturated fatty acids, their melting point becomes lower, and it becomes difficult to crystallize them. Such oil and fat products can be crystallized by another method that does not add trans or saturated fatty acids. Note that hydrogenation technology is no longer used to modify the hardness of oils and fats except when they are fully hydrogenated. The contents of saturated and trans fatty acids can be decreased by introducing MC. Because MC can be used to crystallize oil and fat products, we herein examined the possibility of using MC at the industrial scale.

Previous studies have revealed that when the cooling rate is 15 °C min^−1^, OPO and POP form MC crystals with a composition ratio of 1:1 [7]. In contrast, when rapid cooling is performed at ≥40 °C min^−1^, OPO and POP form eutectic crystals, instead of an MC phase [4]. Therefore, the purpose of the present study is to clarify the mechanism of MC formation under various cooling conditions. We also clarified the effect of changing the composition of OPO and POP on MC formation, because the raw oils and fats change their TAG composition depending on the season and production area [1]. The crystallization behavior of MC was investigated by differential scanning calorimetry (DSC) and synchrotron radiation time-resolved X-ray diffraction (SR-TXRD) methods, while changing the proportion of OPO and POP from 4:6 to 6:4 (*w*/*w*) and the cooling rate. The molecular weights of POP and OPO change by just under 3%, so the weight ratio and molar ratio take the same number. Therefore, the proportion of OPO and POP from 4:6 to 6:4 (*w*/*w*) = 4:6 to 6:4 (mol/mol). Then, in the following sentences, (mol/mol) will be used instead of (*w/w*).

## 2. Results

### 2.1. Polymorphic Behavior with a Binary Mixture of OPO/POP = 5/5

To clarify the polymorphic behavior using a mixture of OPO/POP = 5/5 (mol/mol), the SR-TXRD measurement was performed at the cooling rates of 5, 10, 15, 20, 25, 30, 35, and 40 °C min^−1^. The SR-TXRD data were shown at the typical cooling rates of 20, 25, and 35 °C min^−1^ (Figure 1). All results of the SR-TXRD measurement were magnified to observe the weak diffraction peaks.

Figure 1a,b depicts the polymorphic behavior of a mixture of OPO/POP = 5/5 when cooled from 100 °C to −50 °C at a cooling rate of 20 °C min^−1^. We observed diffraction peaks of 4.7 and 0.42 nm (open triangles in Figure 1a,b) at −2 °C, which correspond to the double-chain-length structure and α form of POP (α-2 POP). Diffraction peaks of 4.3, 0.45, and 0.39 nm (closed triangles depicted in Figure 1a,b) at −9 °C correspond to the double-chain-length structure and β form of MC (β-2 MC). The diffraction patterns observed when the cooling rate was lower than 20 °C min^−1^ (data not shown) were the same as those obtained at a cooling rate of 20 °C min^−1^. This result indicates that the MC should be chiefly formed after performing POP crystallization at a cooling rate of ≤20 °C min^−1^, as reported elsewhere [4,7,14].

According to the SR-TXRD result obtained at a cooling rate of 25 °C min^−1^ (Figure 1c,d), the diffraction peaks of 4.7 and 0.42 nm (arrows in Figure 1c,d) derived from α-2 POP were observed at −5 °C. Slight diffraction peaks of 5.2, 2.61, 0.416, and 0.378 nm (asterisks in Figure 1c,d) were observed at −8 °C, corresponding to the double-chain-length and sub-α form of POP (sub-α-2 POP). We also observed diffraction peaks of 4.3 and 0.45 nm (closed triangles in Figure 1c,d) derived from β-2 MC at −11 °C. Diffraction peaks of 2.56 and 0.412 nm (open arrows in Figure 1c,d) were observed at −26 °C, and they correspond to the double-chain-length and α form of OPO (α-2 OPO). The diffraction peak of 2.56 nm is the second-order diffraction peak of 5.3 nm derived from α-2 OPO. In addition, the diffraction peak of 0.378 nm (asterisk in Figure 1d) gradually shifted to 0.366 nm with an increase in temperature. This shift indicates that the reduction in molecular motion caused by cooling decreased the short spacing.

These results show highly complex phase behavior of the binary mixture, because some OPO and POP must crystallize separately, while the others must form MC crystals. Previous studies have shown that the α phase exhibits a reversible transition to the sub-α phase [19,20]. Although the sub-α phase of POP was observed in few previous reports [4,21,22,23], its structure has not been clarified. Thus, the structure of the sub-α phase is discussed later in the paper. A comparison of the diffraction peak pattern obtained with each cooling rate showed that the diffraction pattern at a cooling rate of 30 °C min^−1^ (data not shown) was the same as that obtained at 20 °C min^−1^. However, the diffraction intensity at 4.3 nm (β-2 MC) was weaker, while it was stronger at 5.3 nm (sub-α-2 POP) than those observed at 20 °C min^−1^.

At the cooling rate of 35 °C min^−1^ (Figure 1e,f), the diffraction peaks of 5.3, 0.416, and 0.378 nm (denoted by arrows) derived from sub-α-2 POP at −8 °C and the diffraction peaks of 2.56 and 0.412 nm (denoted by open triangles) derived from α-2 OPO at −30 °C were observed. The broad diffraction peak of 3.2 nm (diamond in Figure 1e) derived from OPO crystals was observed and its intensity gradually increased with decreasing temperature. In contrast, the diffraction peak of 4.7 or 4.3 nm was not observed (Figure 1a–d). In addition, the diffraction pattern at a cooling rate of 40 °C min^−1^ (data not shown) was the same as that observed at 35 °C min^−1^, suggesting that MC crystals must form at cooling rates lower than 35 °C min^−1^.

### 2.2. Polymorphic Behavior with a Binary Mixture of OPO/POP = 4/6

We conducted a DSC cooling experiment using a binary mixture of OPO/POP = 4/6 (mol/mol). Figure 2 shows the cooling thermograms. The sample was cooled from 100 °C to −50 °C at the cooling rates of 1, 5, 10, 15, 20, 25, 30, 35, 40, 50, 100, and 150 °C min^−1^. The peak top temperatures are summarized in Figure 6.

In addition, to clarify the polymorphic behavior using the mixture of OPO/POP = 4/6, the SR-TXRD measurement was performed at various cooling rates (Figure 1). The SR-TXRD data were shown at the typical cooling rates of 5, 30, and 35 °C min^−1^ (Figure 3).

When the DSC cooling thermogram was obtained at a cooling rate of 1 °C min^−1^ (Figure 2), we observed two exothermic peaks (black arrow and closed triangle). When the cooling rate was increased to 5 °C min^−1^, both peaks were observed at a lower temperature.

At the cooling rate of 5 °C min^−1^, diffraction peaks of 4.7 and 0.42 nm (indicated by arrows) derived from α-2 POP at 4 °C were obtained (SR-TXRD result, Figure 3a,b). The diffraction peaks of 4.3 and 0.45 nm (closed triangles) derived from β-2 MC appeared at 0 °C and that of 6.2 nm (open triangle in Figure 3a) were observed at −5 °C. The peak will correspond to the triple-chain-length and β form of OPO (β-3 OPO). No exothermic peak corresponding to the diffraction peak of 6.2 nm was observed (DSC result, Figure 2), probably because the sample mass used in DSC is approximately one-tenth of that used in SR-TXRD.

The exothermic peak denoted by arrows in Figure 2 must correspond to α-2 POP crystallization, while those denoted by closed triangles must correspond to β-2 MC crystallization at a cooling rate of 5–40 °C min^−1^, according to the SR-TXRD result shown in Figure 3 and Table 1. However, the on-set temperature of the exothermic peak denoted by the arrow at a cooling rate of 1 °C min^−1^ is much higher than that at 5 °C min^−1^. Therefore, the POP would form a more stable polymorph than α-2 POP. Similarly, the SR-TXRD result showed a diffraction peak of 6.2 nm derived from β-3 OPO when a mixture of OPO/POP = 5/5 was used (Figure 3a).

The SR-TXRD result showed that the diffraction pattern obtained at a cooling rate of 10 °C min^−1^ (data not shown) was almost the same as that obtained at 5 °C min^−1^, except that the diffraction peak of 6.2 nm (β-3 OPO) was not observed. Similar results were obtained till the cooling rate was 30 °C min^−1^, except for the difference in their intensity (Figure 3c,d). The diffraction intensities of 4.7 and 0.42 nm (α-2 POP) gradually increased and those of 4.3 and 0.45 nm (β-2 MC) gradually decreased when the cooling rate was increased (Figure 3a–f).

The DSC result obtained at a cooling rate of 30 °C min^−1^ (Figure 2) showed that the MC crystallization peak (closed triangle) had a drastically decreased peak area. As the cooling rate was increased to ≥50 °C min^−1^, the exothermic peaks denoted by closed triangles in Figure 2 disappeared, as pointed out in our previous work [4]. In contrast, we observed the exothermic peak denoted by an arrow in Figure 2 even when the cooling rate was increased to 150 °C min^−1^. In addition, we observed new exothermic peaks (denoted by open triangles) at cooling rates of ≥25 °C min^−1^.

The SR-TXRD result obtained at a cooling rate of 35 °C min^−1^ (Figure 3e,f) showed that the diffraction peaks of 4.7 and 0.42 nm derived from α-2 POP were detected at 4 °C. At −18 °C, slight diffraction peaks of 4.3 and 0.45 nm derived from β-2 MC were observed. Similar results were obtained at a cooling rate of 40 °C min^−1^ (data not shown). Although three exothermic peaks were observed in the DSC result, no diffraction peak corresponding to the new exothermic peak was noted (open triangles in Figure 2).

Hence, MC crystals must be formed at cooling rates of ≤40 °C min^−1^ using the OPO/POP = 4/6 mixture. When the OPO/POP = 4/6 mixture was used, the MC crystals were obtained at a higher cooling rate than those obtained with the OPO/POP = 5/5 mixture (Figure 1 vs. Figure 3).

### 2.3. Polymorphic Behavior with a Binary Mixture of OPO/POP = 6/4

We carried out a cooling experiment using the DSC (Figure 4) and SR-TXRD (Figure 5) with a binary mixture of OPO/POP = 6/4 (mol/mol). The cooling rate and the temperature range shown in Figure 4 and Figure 5, respectively, were the same as those shown in Figure 2 and Figure 3. The peak top temperatures obtained using the DSC are shown in Figure 6. The SR-TXRD data are shown under the typical cooling rates of 10, 15, and 25 °C min^−1^ (Figure 5). Only one exothermic peak was observed when the DSC measurement was carried out at the cooling rates of 1 and 5 °C min^−1^ (Figure 4). When the cooling rate was increased to 10 °C min^−1^, two new small exothermic peaks (arrow and open triangle) were observed.

Figure 5 illustrates the polymorphic behavior measured by the SR-TXRD method using the binary mixture of OPO/POP = 6/4. The diffraction peaks of 4.7 and 0.42 nm (arrows in Figure 5a,b) derived from α-2 POP were detected at −4 °C. The diffraction peaks of 4.3 and 0.45 nm (closed triangles in Figure 5a,b) derived from β-2 MC appeared at −7 °C when the cooling rate of 10 °C min^−1^ was used. The diffraction peak of 6.2 nm (open triangle in Figure 5a) derived from β-3 OPO was observed at −13 °C.

The diffraction pattern obtained using the SR-TXRD method for the OPO/POP = 6/4 mixture at a cooling rate of 10 °C min^−1^ was the same as that obtained at a cooling rate of 5 °C min^−1^ (data not shown), except for the intensity of the diffraction peak of 6.2 nm (β-3 OPO). The intensity of the diffraction peak obtained at the cooling rate of 5 °C min^−1^ was stronger than that at 10 °C min^−1^. Though only one exothermic peak seems to occur at the cooling rate of 5 °C min^−1^ (Figure 4), the crystals coexist in three types of polymorphs because the diffraction pattern at 5 °C min^−1^ was the same as that at 10 °C min^−1^ (Figure 5a). In Figure 4, the two exothermic peaks observed at the cooling rate of 15 °C min^−1^ (arrow and open triangle) had a higher total area than that observed at the cooling rate of 10 °C min^−1^. In contrast, the peak area indicated by the closed triangle decreased and disappeared at the cooling rates of >15 °C min^−1^.

The SR-TXRD result at the cooling rate of 15 °C min^−1^ is shown in Figure 5c,d. The diffraction peaks of 5.3, 2.61, 0.416, and 0.378 nm (indicated by arrows) derived from sub-α-2 POP were first observed at −7 °C. Next, the diffraction peaks of 4.3 and 0.45 nm (indicated by closed triangles) derived from β-2 MC appeared at −14 °C, and the diffraction peaks of 2.56 and 0.412 nm (indicated by open triangles) derived from α-2 OPO were observed at −28 °C. The DSC and SR-TXRD results indicate that β-2 MC begins to disappear, and sub-α-2 POP and α-2 OPO crystals appear separately at the cooling rate of 15 °C min^−1^.

The diffraction pattern at the cooling rate of 15 °C min^−1^ (data not shown) is the same as that obtained at 20 °C min^−1^, except for the diffraction intensity. The diffraction intensity of 5.3 nm (sub-α-2 POP) increased and that of 4.3 nm (β-2 MC) drastically decreased at the cooling rate of 20 °C min^−1^. When the cooling rate was 25 °C min^−1^ (Figure 5e,f), the diffraction peaks of 5.3, 2.61, 0.416, and 0.378 nm (denoted by arrows) derived from sub-α-2 POP were observed at −7 °C. The intensity of the diffraction peak of 3.2 nm (diamond in Figure 5e) derived from OPO crystals gradually increased as the temperature was decreased. In addition, the diffraction peaks of 2.56 and 0.412 nm (open triangles in Figure 5e,f) derived from α-2 OPO were observed at −30 °C. The diffraction patterns remained unchanged at the cooling rates of ≥25 °C min^−1^ (data not shown). The results obtained at the cooling rates of ≥25 °C min^−1^ reveal that β-2 MC must disappear completely and α-2 OPO and sub-α-2 POP must crystallize independently. The cooling rate at which β-2 MC crystals disappeared was lower than that observed with the OPO/POP = 5/5 mixture (Figure 1 vs. Figure 5).

## 3. Discussion

Figure 6 presents a comparison of crystallization temperature using the binary mixtures of OPO/POP = 4/6, 5/5, and 6/4 for all DSC results. The crystallization temperature of POP decreased, and hence, the degree of POP supercooling increased when the proportion of POP was decreased in the binary mixture. In contrast, no significant change was observed in the crystallization temperatures of OPO and MC.

In addition, we have summarized the polymorphs observed in all SR-TXRD results using the binary mixtures of OPO/POP = 4/6, 5/5, and 6/4 in Table 2. Moreover, MC crystals drastically disappeared and the binary system formed a eutectic phase when the POP formed sub-α. The disappearance of the MC crystals can be explained by the increase in the POP crystallization rate when the α-form is transformed to the sub-α form and POP crystalizes independently. In our previous study [4], MC formation stops at a cooling rate of >40 °C min^−1^. The results shown in Figure 1, Figure 2, Figure 3, Figure 4, Figure 5 and Figure 6 and Table 2 strongly support the result of our previous study.

The chain length of sub-α-2 POP was quite long at 5.3 nm. According to the estimated schematic image shown in Figure 7, a stable glycerol conformation is obtained possibly owing to the formation of sub-α-2 POP. The sub-α form structure has not been much studied except for the Yano et al. study, according to which a subcell structure forms in 1,3-distearoyl-2-oreoyl-*sn*-glycerol [20]. Thus, further analysis is needed to understand the mechanisms of the stabilization of sub-α-2 POP.

Another possible reason for the disappearance of MC crystals can be the structural difference between MC and POP. Sub-α-2 POP formed a 5.3-nm-long chain, which is different from that of β-2 MC of 4.3 nm. This structural difference can be responsible for the decrease in the stability of MC crystals because of a disorder of methyl end stacking, a steric hindrance of glycerol conformation, and a decrease in molecular interactions between aliphatic chains [24].

Figure 7 illustrates schematic images of α-2 POP, β-2 MC, and sub-α-2 POP. We estimated the values of the chain-length and subcell structures of POP and MC by using the values of the carbon–carbon distance, carbon–oxygen distance, palmitoyl moiety length, and oleoyl moiety length at approximately 0.15, 0.14, 2.2, and 2.0 nm, respectively [25]. In addition, we compared the estimated schematic images with the long and short spacings obtained in this study.

The estimated length of α-2 POP nearly coincided with the long and short spacings. The chain length of β-2 MC was shorter than that of α-2 POP. Hence, it is necessary to tilt and increase the density of the aliphatic chains to conform to the chain-length and the subcell structure. The MC structure indicates that it would be stabilized according to the following three indices: molecular interactions of the aliphatic chains, methyl end stacking, and glycerol conformation. Palmitoyl and oleoyl chains are located in the same leaflet of β-2 MC; as a result, the methyl end stacking is parallel to the lamellar plane. Furthermore, OPO and POP have the same char-like glycerol conformation. Therefore, it is considered that MC crystallizes in a short time despite being a stable polymorph and the diffraction peaks of MC became very sharp [4]. Consequently, the schematic images of α-2 POP and β-2 MC were similar to those shown in previous studies [7,9].

The POP crystals formed first and we found the coexistence of MC and POP (see Table 2). In addition, the crystallinity was reduced because the diffraction peak derived from the β form observed from the cooling measurement in this study was broader than that observed from heating after cooling measurement in the previous study [4]. Therefore, it is necessary to conduct transformation by heating in order to form MC crystals with high crystallinity. While manufacturing oil and fat products, the cooling temperature can be set for each product separately. We will investigate the crystallization behavior of MC at the set temperature in a future study.

In conclusion, we showed that increasing the cooling rate destabilized the MC formation. When the proportion of OPO is increased, MC formation is destabilized. In contrast, increasing the proportion of POP stabilized MC formation under rapid cooling conditions. Hence, it is clear that the stability of polymorphism can be controlled by changing the composition ratio of OPO and POP.

## 4. Materials and Methods

### 4.1. Materials

TAGs of OPO and POP with the purity of >99% were purchased from Tsukishima Foods Industry Co., Ltd. (Tokyo, Japan) and used without further purification. The samples were heated to melting and mixed in a 0.5 mL microcentrifuge tube (Labdhi Enterprise: Gujarat, India).

### 4.2. Measurements

DSC methods were performed with a DSC 8500 instrument (PerkinElmer Inc., Waltham, MA, USA) at atmospheric pressure over the temperature ranging from −50 to 100 °C. Approximately 2 mg of the samples were put into aluminum pans with covers. Dry helium was used as a purge gas and flown into the DSC cell at 20 mL/min. The DSC was cooled by a CLN2 instrument using liquid N_2_. The CLN2 temperature was set at −190 °C. An empty pan was used as a reference. The DSC calibration was carried out by referring to the enthalpy and melting points of cyclohexane and indium standard (melting point of cyclohexane = 6.56 °C, melting point of indium = 156.6 °C; ΔH = 28.45 J/g). The samples were cooled from 100 to −50 °C at the cooling rates of 1, 5, 10, 15, 20, 25, 30, 40, 50, 100, and 150 °C min^−1^. Next, they were heated from −50 to 80 °C at the rate of 10 °C min^−1^. The resulting thermograms were analyzed using Pyris software to calculate the enthalpy and peak top temperature.

The SR-TXRD methods were performed at BL19B2 Synchrotron Radiation Facility SPring-8 at the Japan Synchrotron Radiation Research Institute in Hyogo, Japan. The samples were weighed into aluminum cells (12 mm dia. × 1 mm) with a 5 mm dia. hole covered with a polyimide film.

The temperature program was controlled by LINKAM TH-600 (Linkam Co., Surrey, UK). The sample cells were put in the LINKAM, and the samples were cooled from 100 °C to −50 °C at the cooling rates of 5, 10, 15, 20, 25, 30, 35, and 40 °C min^−1^. The samples were also heated from −50 °C to 80 °C after cooling at the rate of 10 °C min^−1^. In the present SR-TXRD study, the experiment at a cooling rate of >40 °C min^−1^ was not performed because of a limitation of cooling capacity. The energy of the incident X-rays was 24 keV (wave length ~0.05 nm). X-ray scattering data were collected using PILATUS 2M semiconducting detector (DECTRIS Ltd., Baden-Daettwil, Switzerland).

The samples were placed from the detector to a 741 mm camera length upstream. The camera length was calibrated with a diffraction pattern of tripalmitin and silver behenate. Each sample was exposed for 1 s at an interval of 2 s. The data were transferred in 0.33 s. The diffraction intensity was standardized by the incident X-ray intensity before the sample. The measurement was performed three times for each sample, and the representative data are shown. The obtained concentric data were made one-dimensional by integrating and averaging in the circumferential direction using Plot Radially 2.2, a plug-in of Image J (1.50i).

## Figures and Tables

**Figure 1 molecules-25-05253-f001:**
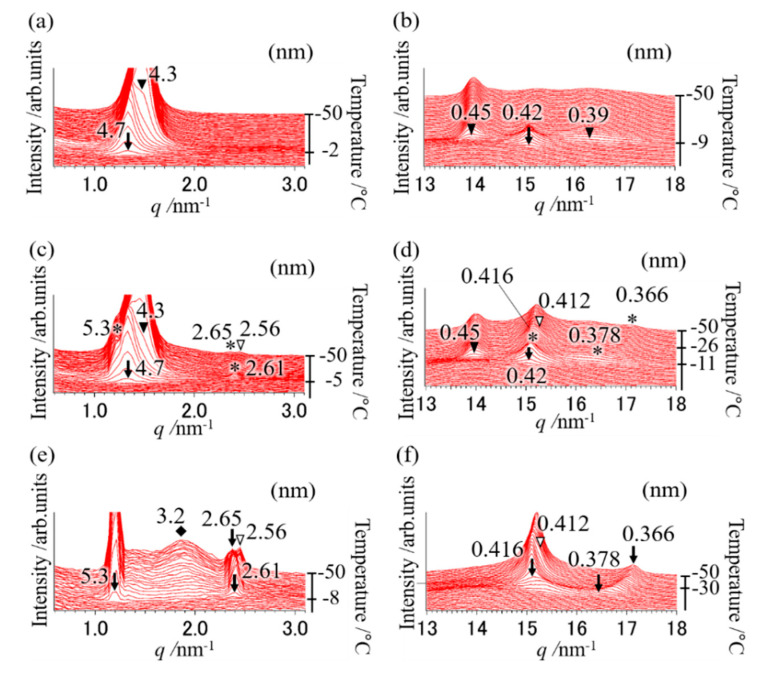
Synchrotron radiation time-resolved X-ray diffraction (SR-TXRD) results at the cooling rate of (**a**,**b**) 20 °C min^−1^, (**c**,**d**) 25 °C min^−1^, and (**e**,**f**) 35 °C min^−1^ with a binary mixture of OPO/POP = 5/5. The x-axes show q value (q=4π·sin(θ)/λ). The left y-axes show diffraction intensity. The right y-axes show temperature. (**a**,**c**,**e**) indicate magnification of small-angle pattern. (**b**,**d**,**f**) indicate magnification of wide-angle pattern. The arrows and asterisks are derived from POP crystals (4.7 and 0.42 nm: α form, 5.3, 0.416, etc., nm: sub-α form). The closed triangles are derived from β-2 MC. The open triangles and diamond are derived from α-2 OPO.

**Figure 2 molecules-25-05253-f002:**
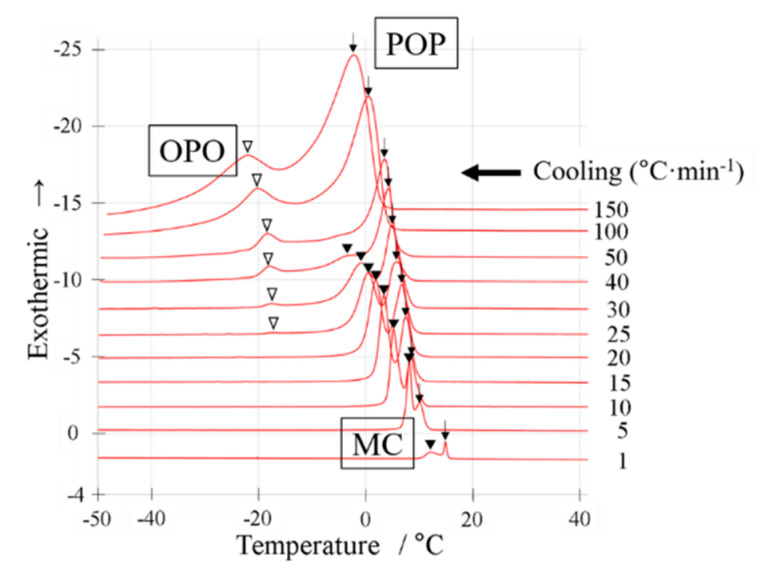
Differential scanning calorimetry (DSC) cooling thermograms with a binary mixture of OPO/POP = 4/6. The x-axis shows temperature. The y-axis shows normalized exothermic heat flow. The thermograms overlay at cooling rate of 1~150 °C min^−1^ from bottom to top. The open triangles indicate OPO crystallization. The closed triangles indicate MC crystallization. The arrows indicate POP crystallization.

**Figure 3 molecules-25-05253-f003:**
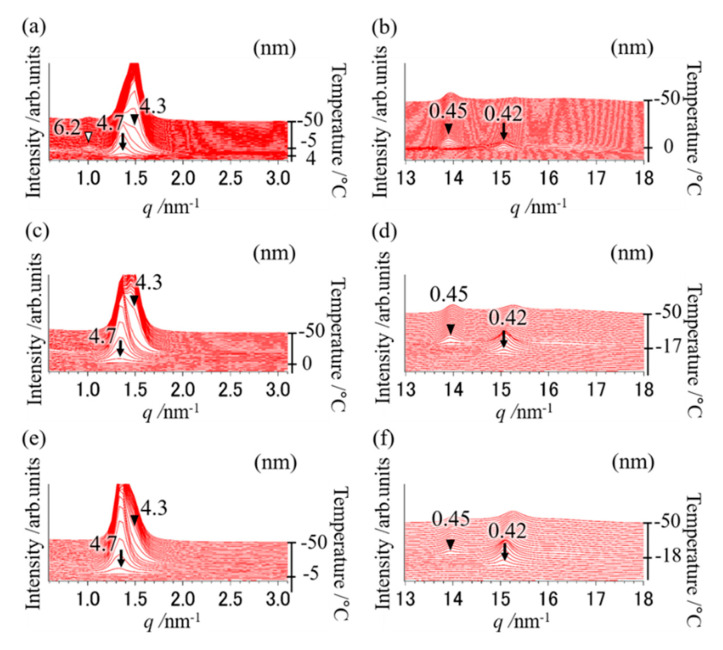
SR-TXRD results at the cooling rate of (**a**,**b**) 5 °C min^−1^, (**c**,**d**) 30 °C min^−1^, and (**e**,**f**) 35 °C min^−1^ with a binary mixture of OPO/POP = 4/6. The x-axes show q value (q=4π·sin(θ)/λ). The left y-axes show diffraction intensity. The right y-axes show temperature. (**a**,**c**,**e**) indicate magnification of small-angle pattern. (**b**,**d**,**f**) indicate magnification of wide-angle pattern. The arrows are derived from α-2 POP. The closed triangles are derived from β-2 MC. The open triangle is derived from β-3 OPO.

**Figure 4 molecules-25-05253-f004:**
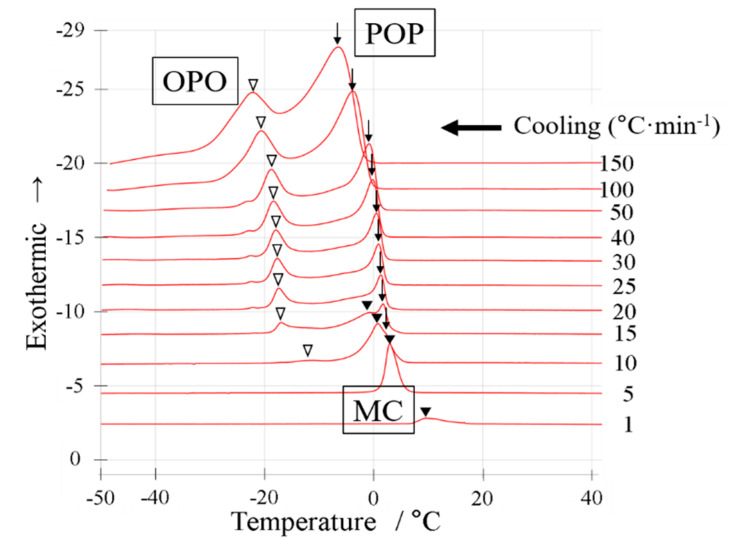
DSC cooling thermograms with a binary mixture of OPO/POP = 6/4. The x-axis shows temperature. The y-axis shows normalized exothermic heat flow. The thermograms overlay at cooling rate of 1~150 °C·min^−1^ from bottom to top. The open triangles indicate OPO crystallization. The arrows indicate POP crystallization. The closed triangles indicate MC crystallization.

**Figure 5 molecules-25-05253-f005:**
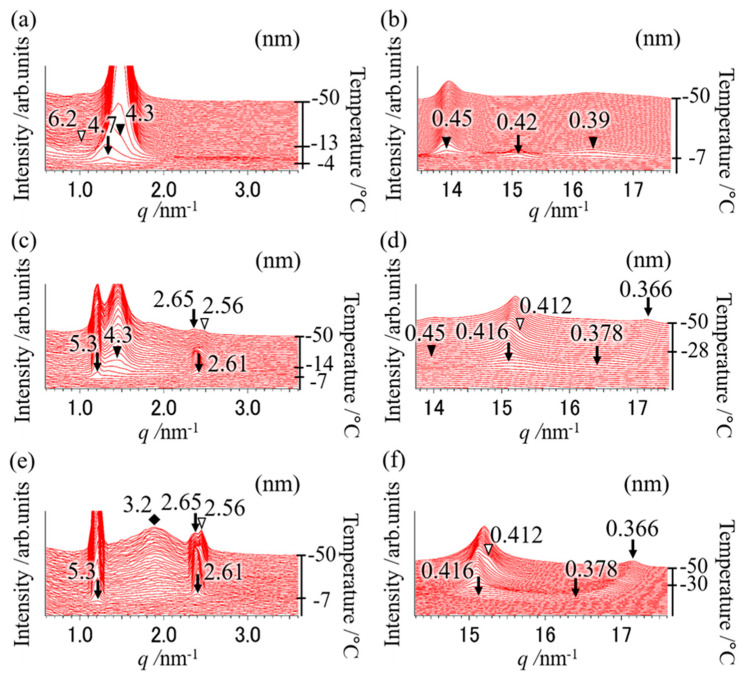
SR-TXRD results at the cooling rate of (**a**,**b**) 10 °C min^−1^, (**c**,**d**) 15 °C min^−1^, and (**e**,**f**) 25 °C min^−1^ with a binary mixture of OPO/POP = 6/4. The x-axes show q value (
q=4π·sin(θ)/λ). The left y-axes show diffraction intensity. The right y-axes show temperature. (**a**,**c**,**e**) indicate magnification of small-angle pattern. (**b**,**d**,**f**) indicate magnification of wide-angle pattern. The arrows are derived from POP crystals (4.7 and 0.42 nm: α form, 5.3, 0.416, etc., nm: sub-α form). The closed triangles are derived from β-2 MC. The open triangles and diamond are derived from α-2 OPO.

**Figure 6 molecules-25-05253-f006:**
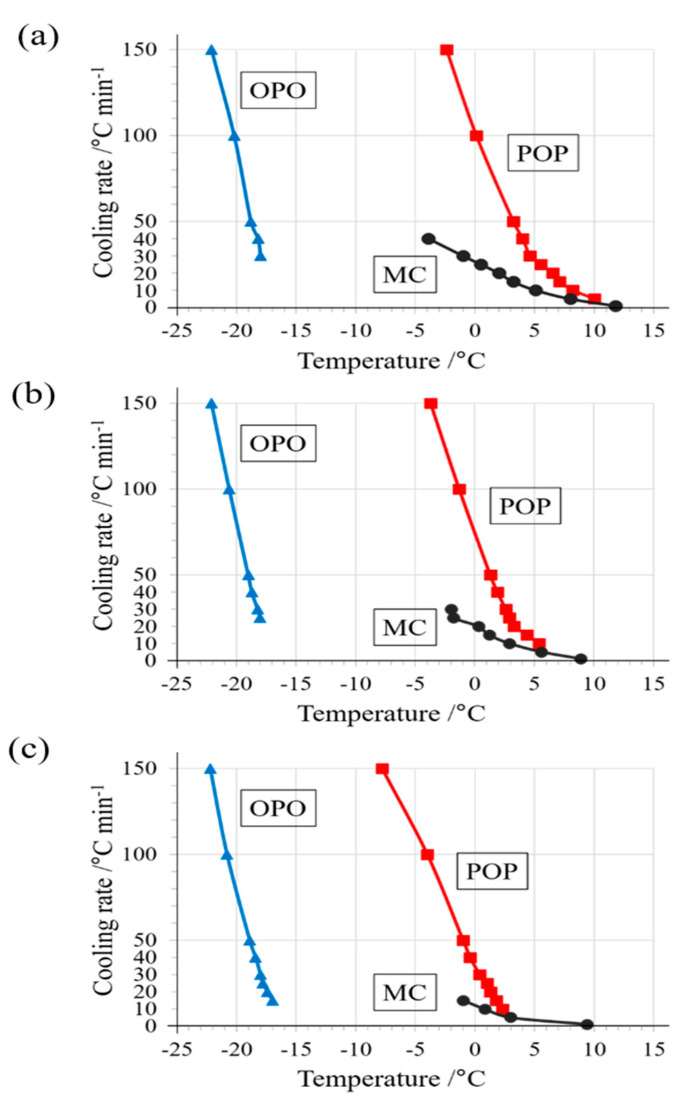
Crystallization temperatures of polymorphs based on DSC exothermic peak top temperatures. Comparing the results of a binary mixture of (**a**) OPO/POP = 4/6, (**b**) 5/5, and (**c**) 6/4. The values of the temperature are represented by circles for MC, red squares for POP, and blue triangles for OPO.

**Figure 7 molecules-25-05253-f007:**
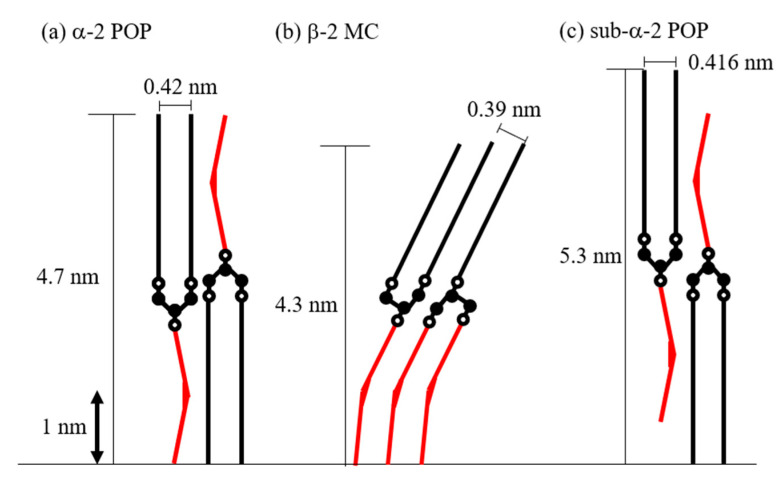
Schematic image of POP and MC structure models. (a) Structure model of α form of POP (α-2 POP) inferred from diffraction peaks of 4.7 and 0.42 nm. (b) Structure model of β form of MC consisted of POP and OPO (β-2 MC) inferred from diffraction peaks of 4.3 and 0.39 nm. (c) Structure model of sub-α form of POP (sub-α-2 POP) inferred from diffraction peaks of 5.3 and 0.416 nm. Closed circle: carbon atom. Open circle: oxygen atom in glycerol backbone. Straight line: palmitoyl moiety. Red bent line: oleoyl moiety.

**Table 1 molecules-25-05253-t001:** Long and short spacing values of molecular compound (MC)_OPO/POP_, 1,3-dioleoyl-2-palmitoyl-*sn*-glycerol (OPO), and 1,3-dipalmitoyl-2-oleoyl-*sn*-glycerol (POP).

MC_OPO/POP_ (nm)						
	α			β		
Long spacing	4.7			4.24		
Short spacing	0.42			0.46		
				0.38		
				0.377		
**OPO**						
	α *	α **	α	β’	β_2_	β_1_
Long spacing	5.4, 5.2	5.1, 3.2	5.3	4.4	5.6	6.7
Short spacing	0.42	0.41	0.42	0.43	-	0.48
				0.44		0.38
						0.37
**POP**						
	sub-α	α	γ	β’	β_2_	β_1_
Long spacing	5.2	4.65	6.54	4.24	6.1	6.1
Short spacing	0.41	0.421	0.474	0.423	0.461	0.461
	0.38		0.39	0.396	0.393	0.372
					0.372	0.367

-: not detected. *: chilled at 15 °C min^−1^. **: chilled at 40 °C min^−1^. Data based on previous works [4,7,8,9,10].

**Table 2 molecules-25-05253-t002:** Polymorphs observed at SR-TXRD results.

OPO/POP = 4/6					
Cooling Rate (°C·min^−1^)	Sub-α-2 POP	α-2 POP	β-2 MC	α-2 OPO	β-3 OPO
5	-	+	+	-	+
10	-	+	+	-	-
15	-	+	+	-	-
20	-	+	+	-	-
25	-	+	+	-	-
30	-	+	+	-	-
40	-	+	+	-	-
**OPO/POP = 5/5**					
Cooling Rate (°C·min^−1^)	Sub-α-2 POP	α-2 POP	β-2 MC	α-2 OPO	β-3 OPO
5	-	+	+	-	-
10	-	+	+	-	-
15	-	+	+	-	-
20	-	+	+	-	-
25	+	+	+	+	-
30	+	+	+	+	-
35	+	-	-	+	-
40	+	-	-	+	-
**OPO/POP = 6/4**					
Cooling Rate (°C·min^−1^)	Sub-α-2 POP	α-2 POP	β-2 MC	α-2 OPO	β-3 OPO
5	-	+	+	-	+
10	-	+	+	-	+
15	+	+	+	+	-
20	+	-	+	+	-
25	+	-	-	+	-
30	+	-	-	+	-
35	+	-	-	+	-
40	+	-	-	+	-

+: existence of polymorph, -: not detected.

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
