# Peer review of "Mixing Ratio and Cooling Rate Dependence of Molecular Compound Formation in OPO/POP Binary Mixture"

_molecules, 2020, doi:10.3390/molecules25225253_

Round 1

Reviewer 1 Report

In the article entitled: “Mixing ratio and cooling rate dependence of molecular compound formation in OPO/POP binary mixture” the Authors described mechanism of MC formation under various cooling conditions, the effect of changing the composition of OPO and POP on MC formation,  the crystallization behaviour of MC using by differential scanning calorimetry and synchrotron radiation time-resolved X-ray diffraction methods after changing the proportion of OPO and POP from 4:6 to 6:4 and the cooling rate. The Authors showed that increasing the cooling rate destabilized the MC formation. In the situation when the proportion of OPO is increased, MC formation is destabilized, whereas increasing the proportion of POP stabilized MC formation under rapid cooling conditions.

Article is written well and will be interesting for a wide range of readers.

In my opinion article is worth publication in the Molecules after minor revision.

Specific comments:

  • Please improve the quality of some Figures (1,3, 5, 6 and 7) and Table 2.
  • Figure 1 and 3 – please describe both axes in the Figure caption; in my opinion the symbol “(nm)” should be putted with all values.
  • Figure 2 – Please describe the symbols of OPO, MC and POP into Figure (for example see Ref. 13) and indicate the cooling direction. Please also change the unit form “0C/min” to “0C*min-1”.
  • Figure 7 – I suggest using different colours instead of straight line for palmitoyl moiety and bent line for oleoyl moiety (for clarity).
  • References – Some of references cited in manuscript are presented in wrong format (e.g. Ref. 22: Appl. Cryst., 2007, 40, s297-s302.); please check it carefully and change.

Author Response

Answer to the Reviewer 1

Thank you for your comments. We will answer to your comments and/or questions as follows;

Specific comments:

  • Please improve the quality of some Figures (1,3, 5, 6 and 7) and Table 2.

In addition to Fig. 1,3,5,6,7, which was pointed out, Fig. 2and 4 and was added to enlarge it for easier viewing.

  • Figure 1 and 3 – please describe both axes in the Figure caption; in my opinion the symbol “(nm)” should be putted with all values.

(answer) The vertical and horizontal axes have been added to the captions in Figures 1 and 3. In addition, although it was pointed out that about "nm", we avoided writing it in all the numerical values in the figure because the figure becomes complicated. Please understand.

  • Figure 2 – Please describe the symbols of OPO, MC and POP into Figure (for example see Ref. 13) and indicate the cooling direction. Please also change the unit form “0C/min” to “0C*min-1”.

(answer) In Fig. 2, as pointed out, POP, MC, and OPO are added next to each symbol, and the cooling direction is indicated by an arrow.

  • Figure 7 – I suggest using different colours instead of straight line for palmitoyl moiety and bent line for oleoyl moiety (for clarity).

(answer) As pointed out, in Figure 7 we used different colours instead of straight line for palmitoyl moiety and bent line for oleoyl moiety.

  • References – Some of references cited in manuscript are presented in wrong format (e.g. Ref. 22:  Cryst., 2007, 40, s297-s302.); please check it carefully and change.

(answer) The corrected References parts in the revised article, see the attached file, are shown in red.

Furthermore, In Figure 6, the colors of the OPO, MC, and POP curves have been changed to make the lines clearer.

That’s all,

Reviewer 2 Report

This is another excellent paper by this group.  This group has shown and proven over the years the formation of a specific molecular compound (solid state solution with a specific stoichiometry) between complementary triglyceride molecules such as POP and OPO, but also SOS and OSO.  It is a universal effect. They are world-experts and pioneers in synchrotron XRD as well a micro-XRD. Here they continue their discovery work. Until now, a major mystery existed on the conditions necessary for the formation of this molecular compound. Here the authors clearly define the cooling rate effects on the crystallization of the three species, alpha-2 POP, beta-2-MC and alpha-2-OPO.  They discover the cooling rate window for the formation of the desirable MC. The MC is desirable since it has a higher solid fat content than the sum of the contributions of its components and this could be advantageous to decrease the total required saturated fatty acids in a commercial fat.  This has obvious health, sustainability and commercial benefits.  This works defines the exact polymorphism and dynamics of polymorphic transformations of the three species, the two components and the MC, as a function of cooling rate.  This information is critical for the manufacture of MC-containing products. As always the science is top-notch, the writing is clear and understandable and I only recommend a few minor edits:

line 101......MC (beta-2 MC?). this is written as (beta-2-POP), which must be a mistake?

Figure 2 legend.....place the symbol right after the description. Do not list all the symbols and then list the description in sequence after. One next to the other please.

Why not refer to the mixtures of OPO and POP in mol/mol ratios as well? This would help with the mental picture of the mixing. Please include mol/mol, maybe right next to w/w

Figure 6.....is this low quality? What is the resolution of this image? Is it publication quality?

Otherwise, fantastic work!

Author Response

Answer to the Reviewer 2

Thank you for your comments. We will answer to your comments and/or questions as follows;

line 101......MC (beta-2 MC?). this is written as (beta-2-POP), which must be a mistake?

(answer) Thank you for pointing out the mistake. We have changed from MC (a-2 POP) to MC (b-2 POP).

Figure 2 legend.....place the symbol right after the description. Do not list all the symbols and then list the description in sequence after. One next to the other please.

(answer) The description in Figure 2 has been modified as follows, as pointed out by referee. The open triangles indicate OPO crystallization. The closed triangles indicate MC crystallization. The arrows indicate POP crystallization.

Why not refer to the mixtures of OPO and POP in mol/mol ratios as well? This would help with the mental picture of the mixing. Please include mol/mol, maybe right next to w/w

 (answer) You are right! Thank you for your timely suggestion! The molecular weight of POP is around 833, and that of OPO is around 857. The difference in molecular weight is less than 3%. Therefore, it is considered that the values do not change between w / w and mol / mol. Then, add the following statement on line 89 where w / w first appears. “The molecular weights of POP and OPO change by just under 3%, so the weight ratio and molar ratio take the same number. Therefore, the proportion of OPO and POP from 4: 6 to 6: 4 (w / w) = 4: 6 to 6: 4 (mol / mol). Then, in the following sentences, (mol / mol) will be used instead of (w / w).”

Figure 6.....is this low quality? What is the resolution of this image? Is it publication quality?

 (answer) In Figure 6, the colors of the OPO, MC, and POP curves have been changed to make the lines clearer.

That’s all,